biomaterials/materials science/biomedical engineering

bone haemostasis, porous composite sheet, sugar-containing hydroxyapatite, plant-derived polymer

**Author for correspondence:**
Kiyoshi Itatani
e-mail: itatani@sophia.ac.jp

# Fabrication of novel bone haemostasis sheet by using sugar-containing hydroxyapatite and plant-derived polymer

Yeonjeong Noh[1], Tomohiro Umeda[1], Yoshiro Musha[2] and Kiyoshi Itatani[1]

[1]Department of Materials and Life Sciences, Sophia University, 7-1 Kioi-cho, Chiyoda-ku, Tokyo 102-8554, Japan
[2]Department of Orthopaedic Surgery, Toho University, 2-17-6 Ohashi, Meguro-ku, Tokyo 153-8515, Japan

YN, 0000-0003-1453-8840; TU, 0000-0003-3788-5132;
YM, 0000-0002-4608-9982; KI, 0000-0001-9422-3165

The fabrication conditions of bone-haemostasis sheet were examined by using (i) phosphoryl oligosaccharides of calcium (POs-Ca), sugar-containing hydroxyapatite ($s$-$Ca_{10}(PO_4)_6(OH)_2$: $s$-HAp) derived from POs-Ca and (ii) natural plant-derived polymers (locust bean gum (LBG), guar gum (GG) and alginate (AG)). The sol, which had been prepared by dissolving 2 mass% LBG/GG and 2 mass% AG into 200 cm$^3$ deionized water and then by agitating at the speed of 20 000 r.p.m., was immersed into 3 mass% POs-Ca solution at room temperature for 24 h; it was hydrothermally treated at 100°C for 5 h, and then freeze-dried at −50°C for 24 h to form porous composite sheet. The microscopic observation showed that the pore sizes were controlled in the range of 5–100 µm by the optimization of LBG/GG ratio. The composite sheet showed the noted uptake of simulated body fluid (1426%) at 37.0°C and also the human blood. Thus, the porous composite sheet was found to be a promising candidate of the bone haemostasis, on the basis of the data of haemostasis, uptake ability of SBF and solubility in acetic acid–sodium acetate buffer solution.

## 1. Introduction

When the bones are cut during the surgery operation, large amounts of bleeding occur from the bone's vessel. Thus, the rapid haemostatic treatment is essential to stop bleeding for the surgery operation. The typical bone haemostasis materials are animal-derived materials,

e.g. beeswax, and the problem exists with regard to the risks of infections and bone healing. Apart from such animal-derived materials, the synthetic alkylene oxide copolymers, which readily achieve haemostasis and do not inhibit bone healing, have started to be commercially available [1]. This material is hydrophilic and water soluble, but is ended up eliminating from the body in the short period of time. The present authors, therefore, paid attention to the novel plant-derived haemostasis materials having bio-absorbability, bone healing (bone regeneration) ability and no risks of infections.

One of the promising plant-derived and bioresorbable biomaterials is phosphoryl oligosaccharides of calcium (POs-Ca) with $Ca^{2+}$ and $PO_4^{3-}$ groups extracted from the potato starch hydrolysate [2]. Such POs-Ca may be partially hydrolysed to form POs-Ca-derived organic compounds and hydroxyapatite ($Ca_{10}(PO_4)_6(OH)_2$; HAp), i.e. sugar-containing HAp (s-HAp), which helps assisting the bone regeneration [2,3]. In order to fully bring out the bone healing/regeneration performance of POs-Ca/s-HAp, we designed the novel plant-derived composite sheet through the combination of POs-Ca or s-HAp with locust bean gum (LBG) and guar gum (GG). The difference in chemical and mechanical properties exists between LBG and GG, depending upon the difference in mannose (M)/galactose (G) ratio [4]. Relating to the LBG (M/G ratio: typically 4.0), for example, the faster absorption of water and swelling is realized by the excellent water absorbing capacity, water uptake and mechanical strength [5]. Although these properties seem to satisfy the conditions of haemostasis, the heating is required for the complete dissolution of LBG into the water, in contrast to the GG (M/G ratio: typically 2.0) being quickly soluble in cold water [6,7] to show noted adhesiveness [8]. The addition of GG to LBG, therefore, is beneficial to encourage the dissolution and bioresorption into the human body fluid after the haemostasis. These LBG and GG should be used in combination with small amount of biocompatible and hydrophilic alginate (AG) extracted from the brown sea algae for the reinforcement of molecular structure, due to the electrostatic bonding in the presence of $Ca^{2+}$ [9]. In this research, the fabrication conditions of bioresorbable haemostasis materials were examined in order to combine POs-Ca or s-HAp with plant-derived polymers, i.e. LBG, GG and AG.

# 2. Experimental procedure

## 2.1. Fabrication of porous composite sheet

The starting sol (200 cm$^3$) was prepared by dissolving 2 mass% LBG–GG and 2 mass% AG into 200 cm$^3$ deionized water heated at 60°C (LBG–GG/AG). The sol was agitated at 20 000 r.p.m. for 3 min to include the bubbles into them, and then immersed into 3 mass% POs-Ca®50 (Ca content: 5.0 mass%, Oji Cornstarch, Tokyo) solution at room temperature (R.T.) for 24 h to form gel. The resulting gel was further hydrothermally treated at 100°C for 5 h in order to encourage the hydrolysis of POs-Ca to form s-HAp. The resulting materials were further freeze-dried at −50°C for 24 h to fabricate the porous composite sheet.

## 2.2. Evaluation

The phase identification was carried out using an X-ray diffractometer with monochromatic CuK$\alpha$ radiation (XRD; model RINT 2000 V/P, Rigaku Corp., Tokyo, 40 kV and 40 mA), and using an attenuated total reflection Fourier-transform infrared spectrophotometer (ATR/FT-IR; model 8600PC, Shimadzu, Kyoto). Differential thermal analysis (DTA) and thermogravimetry (TG) were conducted using approximately 20 mg of sample powder (Thermo Plus EVO2, Rigaku, Tokyo). The microstructures were observed using a field-emission scanning electron microscope (FE-SEM: model SU-8000, Hitachi, Tokyo). Three-dimensional microstructure observation without fracturing was conducted using a micro-computed tomography apparatus (micro-CT; inspeXio SMX-100T, Shimadzu Corp., Kyoto, 50 kV, 40 μA). The tensile strength of the composite sheet was measured on the basis of strain–stress curve, using a universal testing machine (AGS-G, Shimadzu Corp., Kyoto) with the cross-head speed of 0.5 mm min$^{-1}$. The bioresorbability of composite sheet was examined by immersing it into 0.1 mol dm$^{-3}$ acetic acid–sodium acetate buffer solution at 37.0°C (pH = 5.5). The specimen was immersed into the simulated body fluid (SBF) solution, and the uptake amount of SBF solution was checked. The standard SBF solution was prepared by dissolving appropriate amounts of the chemicals in deionized water, i.e. NaCl, NaHCO$_3$, KCl, K$_2$HPO$_4$·3H$_2$0, MgCl$_2$·6H$_2$0, CaCl$_2$, HCl, Na$_2$SO$_4$ and (CH$_2$OH)$_3$CNH$_2$), using 1 mol dm$^{-3}$ of HCl. The concentrations of inorganic ions in the SBF solution were almost the same as the human blood plasma. The specimen was immersed into the human blood, and the uptake amount of blood and the state of the composite sheet were checked

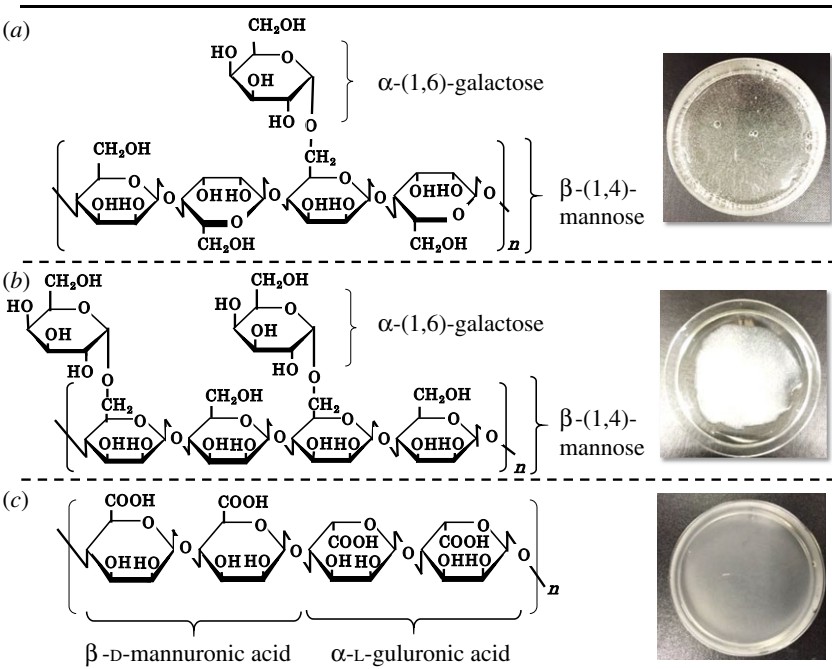

**Figure 1.** Molecular structures of (*a*) LBG, (*b*) GG and (*c*) AG (left side) and photographs of sol formed after the agitation of 2 mass% LBG-, GG- and AG-dissolved solution at 20 000 r.p.m. (3 min) (right side).

after the measurement. Furthermore, the staunching performance was evaluated using porous HAp cube (sizes, $9 \times 9 \times 1\,mm^3$; porosity, 70%; mean pore size, 200 µm) and human blood; the staunching time of blood was defined as the time that the paper on the specimen got wet by the blood coming out on top of the porous HAp cube, due to the capillary force [10]. The fresh human blood was collected from the adult male immediately before the test of haemostasis.

# 3. Results and discussion

## 3.1. Fabrication of porous LBG–GG composite sheet

Prior to checking the fabrication conditions of porous composites by using LBG, GG and AG, the effect of the agitation (20 000 r.p.m.) on the appearances of LBG, GG and AG sol was examined with the results being shown in figure 1, together with the molecular structures. The molecules of both LBG and GG are composed of β-(1,4)-mannose (M) and α-(1,6)-galactose (G) blocks and their M/G ratios are 4.0 and 2.0, respectively. On the other hand, AG consists of randomly arrayed β-D-mannuronic acid and α-L-guluronic acid blocks. Even after the agitation of the sol, the translucencies were kept for LBG and AG sols but the translucency to opacity changes were found for GG sol. Such changes to opacity suggest that the bubbles are included during the agitation of GG sol, and that the viscous properties of sol may preserve the bubbles in the gel. Previously, the viscosities of LBG and GG solutions were examined by Elfak *et al*. [11], who reported that the intrinsic viscosity of GG was higher than that of LBG.

Based upon the fact that the larger amount of bubbles can be included in the GG sol by the agitation, and that the porous composite sheet must be effective for the uptake of larger amount of blood, we firstly examined the fabrication techniques of LBG–GG composites. The microstructures of the composite sheets with different LBG/GG ratios were observed with the results being shown in figure 2. The pore size of the composite sheet was enhanced from 5 to 100 µm with increasing LBG content from 20 to 80 mass% (i.e. with decreasing GG content). The pore size increase with decreasing GG content (i.e. the increase in LBG content) may be explained in terms of the coalescence of pores with decreasing viscous properties.

The tensile strength of the porous LBG/GG composite sheets was measured using the universal testing machine. Typical stress–strain curves of LBG–GG are shown in figure 3. No significant increase in stress was found in the case of 100 mass% GG – 0 mass% LBG sheet (abbreviated as 100GG; figure 3*a*). The stress of 60LBG-40GG composite sheet increased to the maximum (approx.

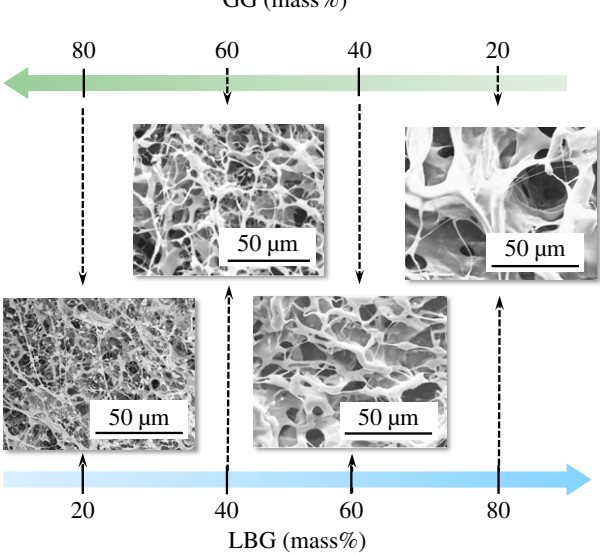

**Figure 2.** FE-SEM micrographs of LBG—GG composite sheets fabricated by agitating the sol at 20 000 r.p.m. (3 min) and then freeze-drying at −50°C for 24 h.

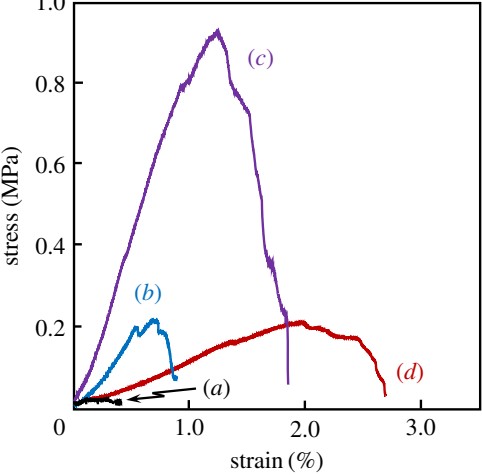

**Figure 3.** Effect of GG/LBG mixing ratio on the tensile stress—strain curve of composite sheet fabricated by agitating sol at 20 000 r.p.m. (3 min) and then freeze-drying at −50°C for 24 h (Cross-head speed: 0.5 mm min⁻¹). (a) 100GG, (b) 60LBG-40GG, (c) 80LBG-20GG and (d) 100LBG.

0.2 MPa) at the strain of 0.7% and then decreased with increasing strain to 0.9% (figure 3b). The stress of 80LBG-20GG composite sheet increased maximum (0.91 MPa) at the strain of around 1.2% and then decreased with increasing strain to 1.9% (figure 3c). On the other hand, the stress of 100LBG sheet increased and attained maximum at 0.2 MPa; on further increase in strain the stress was gradually reduced and fractured at approximately 2.7% (figure 3d).

Based upon the stress–strain curves of LBG–GG composite sheets, the ultimate tensile stresses of LBG–GG composite sheets are plotted against the LBG/GG contents with the results being shown in figure 4. The ultimate tensile stress of LBG/GG composite sheet increased with increasing LBG content (i.e. with decreasing GG content) and showed a maximum at the LBG/GG contents of 80/20 (mass%/mass%). The ultimate tensile stress of 80LBG-20GG composite sheet is higher, compared with the cases of 100LBG and 100GG sheets. Of these materials, GG is known to be the natural adhesive material [8]. The enhanced mechanical properties, due to the combination of LBG and GG, may be explained in terms of the adhesion mechanisms, i.e. mechanical interlocking (chain entanglement), chemical bonding (intermolecular bonding), diffusion and electrostatic bonding [12]. The higher

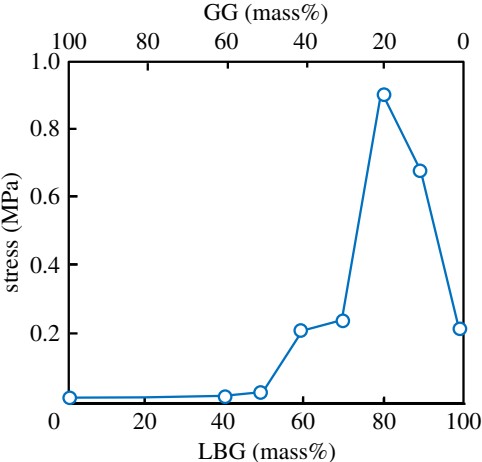

**Figure 4.** Effect of GG/LBG mixing ratio on the ultimate tensile strength of LBG–GG composite sheet fabricated by agitating sol at 20 000 r.p.m. (3 min) and then freeze-drying at $-50°C$ for 24 h.

tensile stress of 80LBG-20GG composite sheet may thus arise from not only these mechanisms, but also total amount of porosity presumably being lowered by the reduced viscous properties.

## 3.2. Fabrication of porous LBG–GG/AG/POs-Ca or s-HAp composite sheet

Since the tensile stress of 80LBG-20GG composite sheet was limited to be below 1 MPa, the small amount of AG (2 mass%) was further added to the starting sol of 80LBG-20GG/AG in order not only to reinforce the mechanical strength but also to encourage the uptakes of the blood as a haemostasis material. For the assistance of the bone regeneration and structure reinforcement, the AG, as well as the POs-Ca or its hydrolysed product (i.e. sugar-containing HAp (s-HAp)), was further added to the 80LBG-20GG/AG composite sheet (i.e. 80LBG-20GG/AG/POs-Ca or 80LBG-20GG/AG/s-HAp, respectively); the cross-linking of calcium ions ($Ca^{2+}$) among AGs may contribute to forming the sponge form [9], showing a haemostatic effect when implanted in a body. Typical XRD patterns of the 80LBG-20GG/AG/POs-Ca and 80LBG-20GG/AG/s-HAp composite sheets are shown in figure 5, together with micro-CT images. XRD patterns showed that no crystalline compound was found from 80LBG-20GG/AG/POs-Ca composite sheet (figure 5a), but that HAp [13] was detected from 80LBG-20GG/AG/s-HAp composite sheet (figure 5b). Micro-CT images showed that the dispersion of inorganic compound (see the red-coloured sites, i.e. HAp) was detected from the 80LBG-20GG/AG/s-HAp composite sheet, in contrast to the case of no crystalline compound being detected from 80LBG-20GG/AG/POs-Ca composite sheet.

The phases present in the composite sheet were checked by ATR/FT-IR spectra with the results being shown in figure 6. FT-IR spectrum of 80LBG-20GG/AG/POs-Ca composite sheet showed that the absorption peaks appeared at 1630, 1415, 1150, 1073, 1021 and 815 $cm^{-1}$ (figure 6a), whereas ATR/FT-IR spectrum of 80LBG-20GG/AG/s-HAp composite sheet indicated that the absorption peaks appeared at 1642, 1428, 1153, 1080, 1022, 936, 880 and 818 $cm^{-1}$ (figure 6b). According to the previous reports of the FT-IR spectra regarding LBG, GG, AG and HAp [14–19], the absorption peaks are assigned to hydroxyl bending, ring stretching of mannose ($>C=O$) and $CO_3^{2-}$ group (1600–1650 $cm^{-1}$), symmetrical deformations of $CH_2$ and COH groups (1400–1430 $cm^{-1}$), C–O vibration modes (1150–1155 $cm^{-1}$), C–O vibration and $CH_2$ twisting modes (1022 $cm^{-1}$), and characteristic absorption peaks of mannose $C_1$–H deformation (880 and 810–820 $cm^{-1}$). Regarding the presence of s-HAp, the absorption peaks at 571 and 610 $cm^{-1}$ are assigned to $v_4$ vibrations of the O–P–O mode, whereas the absorption peaks at 1032 and 1092 $cm^{-1}$ correspond to the $PO_4^{3-}$ functional group. Since the POs-Ca possesses the $Ca^{2+}$ and $PO_4^{3-}$ groups, the formation of HAp may be preferentially occurred by the reaction of $Ca^{2+}$ with phosphate ions (for example, $HPO_4^{2-}$, $H_2PO_4^{-}$, etc.), due to the solubility product being lower, compared with the case of other calcium phosphates [20,21].

The calcium phosphate content (including HAp) in the composite sheet was checked, on the basis of the DTA-TG results being shown in figure 7. DTA curve showed that the endothermic events started to occur at temperature exceeding R.T. and at around 140°C, whereas the exothermic events started to occur at around 260°C and 380°C. The step-wise mass losses occurred, corresponding to the endothermic and exothermic

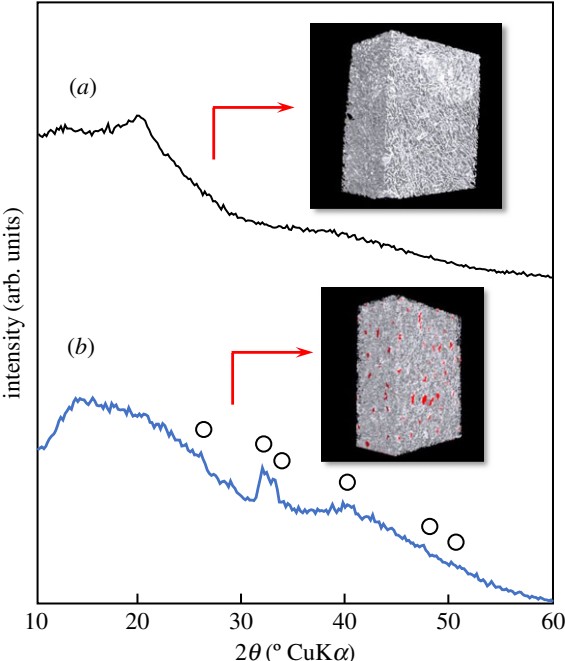

**Figure 5.** XRD patterns of (a) 80LBG-20GG/AG/POs-Ca composite sheet and (b) 80LBG-20GG/AG/s-HAp composite sheet fabricated by agitating sol at 20 000 r.p.m. (3 min), immersing in 3 mass% POs-Ca solution (R.T. for 24 h), hydrothermally treating at 100°C for 5 h (in the case of s-HAp formation), and then freeze-drying at −50°C for 24 h. Open circles: HAp.

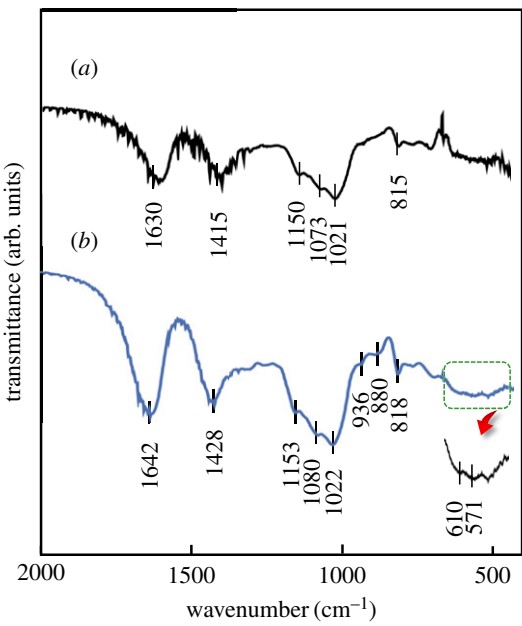

**Figure 6.** FT-IR spectra of (a) 80LBG-20GG/AG/POs-Ca composite sheet and (b) 80LBG-20GG/AG/s-HAp composite sheet by agitating sol at 20 000 r.p.m. (3 min), immersing in 3 mass% POs-Ca solution (R.T. for 24 h), hydrothermally treating at 100°C for 5 h (in the case of s-HAp formation), and then freeze-drying at −50°C for 24 h.

events. The endothermic events at R.T. and 140°C are chiefly attributed to the elimination of physically and chemically adsorbed waters, respectively, whereas the exothermic events at 260°C and 380°C seem to be ascribed to the thermal decomposition of organic compound to form carbon and the oxidation of carbon, respectively. On the basis of the DTA-TG results, the mass decrease in the range of 200°C–600°C seems to correspond to the content of organic materials ((a) in figure 7), and that of the residual mass above 600°C corresponds to the content of calcium phosphate, i.e. HAp ((b) in figure 7).

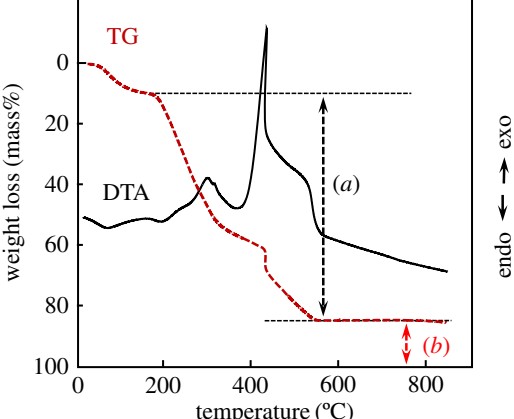

**Figure 7.** Typical DTA-TG curves of 80LBG-20GG/AG/s-HAp composite sheet fabricated by agitating sol at 20 000 r.p.m. (3 min), immersing in 3 mass% POs-Ca solution (R.T. for 24 h), hydrothermally treating at 100°C for 5 h and then freeze-drying at −50°C for 24 h. (a) Organic materials content. (b) Calcium phosphate content. Heating rate: 10°C min$^{-1}$.

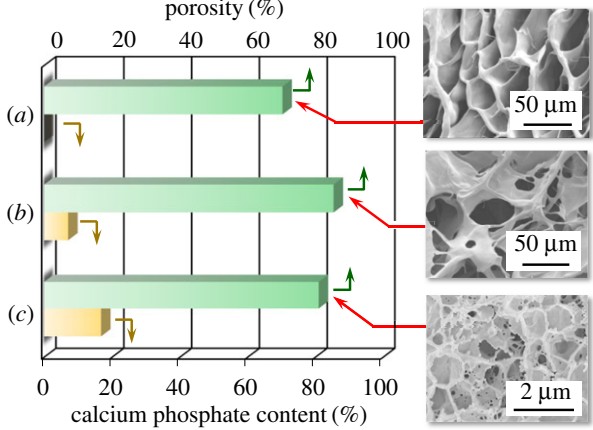

**Figure 8.** Calcium phosphate contents and porosities of (a) 80LBG-20GG/AG composite sheet, (b) 80LBG-20GG/AG/POs-Ca composite sheet and (c) 80LBG-20GG/AG/s-HAp composite sheet fabricated by agitating sol at 20 000 r.p.m. (3 min), immersing in 3 mass% POs-Ca solution (R.T. for 24 h), hydrothermally treating at 100°C for 5 h (except for 80LBG-20GG/AG/ POs-Ca composite), and then freeze-drying at −50°C for 24 h.

The content of calcium phosphate checked by DTA-TG results is shown in figure 8, together with the porosity (checked by micro-CT) and FE-SEM micrographs. FE-SEM micrograph showed that the pores with the sizes of around 50 μm were present in 80LBG-20GG/AG composite sheet (figure 8a), those with the sizes of 10–50 μm in 80LBG-20GG/AG/POs-Ca composite sheet (figure 8b) and those with the sizes of 1–2 μm in 80LBG-20GG/AG/s-HAp composite sheet (figure 8c). Further, the porosities of these composites checked by micro-CT were: 85.4% (80LBG-20GG/AG/POs-Ca composite sheet), greater than 81.0% (80LBG-20GG/AG/s-HAp composite sheet) and greater than 70.3% (80LBG-20GG/AG composite sheet). On the other hand, the calcium phosphate contents checked by DTA-TG were: 16.8 mass% (80LBG-20GG/AG/s-HAp composite sheet), greater than 7.1 mass% (80LBG-20GG/AG/POs-Ca composite sheet) and greater than 0 mass% (80LBG-20GG/AG composite sheet).

The higher porosity of 80LBG-20GG/AG/POs-Ca composite sheet, compared with the case of 80LBG-20GG/AG/s-HAp composite sheet, seems to be attributed to the partial vaporization of POs-Ca during the freeze-drying process. The partial vaporization of POs-Ca, together with the sublimation of the physically adsorbed water during the freeze-drying process, may be supported by the calcium phosphate content in 80LBG-20GG/AG/POs-Ca composite sheet being lower (7.1 mass%), rather than the case of 80LBG-20GG/AG/s-HAp composite (16.8 mass%). The decrease in pore size of composite sheet, due to the addition of POs-Ca and s-HAp to the 80LBG-20GG/AG composite sheet, is explained by assuming that the

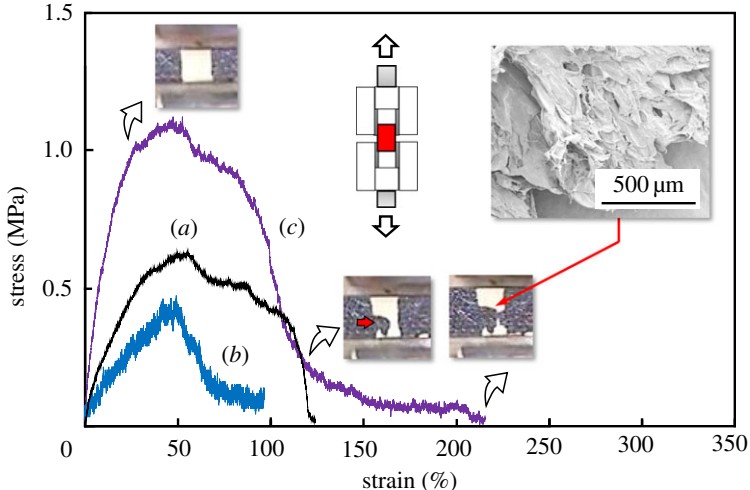

**Figure 9.** Tensile stress–strain curves of (*a*) 80LBG-20GG/AG composite sheet, (*b*) 80LBG-20GG/AG/POs-Ca composite sheet, (*c*) 80LBG-20GG/AG/*s*-HAp composite sheet fabricated by agitating sol at 20 000 r.p.m. (3 min), immersing in 3 mass% POs-Ca solution (R.T. for 24 h), hydrothermally treating at 100°C for 5 h (except for 80LBG-20GG/AG/POs-Ca composite sheet) and freeze-drying at −50°C for 24 h. Note that the photographs showing the fracturing and FE-SEM micrograph are included in the figure.

coalescence of pores may be retarded by the gelation of sol, due to the structural reinforcement by the chemical bridging of calcium ions ($Ca^{2+}$) among AGs.

Figure 9 shows the tensile stress–strain curves of the composite sheets, together with the photographs showing the fracturing of composite sheet and FE-SEM micrograph of the fractured surface. The overall trend revealed that the stress increased with strain and attained maximum; on further increase in strain, the stress was gradually reduced down. The ultimate tensile stresses of composite sheets were: 80LBG-20GG/AG/*s*-HAp composite sheet (1.15 MPa), greater than 80LBG-20GG/AG composite sheet (0.65 MPa) and greater than 80LBG-20GG/AG/POs-Ca composite sheet (0.48 MPa). The noted ultimate tensile stress was found in the case of 80LBG-20GG/AG/*s*-HAp composite sheet. Taking notice of the 80LBG-20GG/AG/*s*-HAp composite sheet, the fracture occurred from left to right side with increasing strain (see photographs). FE-SEM micrograph of the 80LBG-20GG/AG/*s*-HAp composite sheet showed that the complicated fractured surface was observed with increasing strain. The crack initiation and subsequent propagation may be inhibited by the presence of fibres, e.g. (i) restricted crack propagation due to the deflection of crack tips, (ii) formation of bridges across crack faces, and (iii) energy absorption during the pull-out as the fibre debonding from the matrix. Such complicated fracture surface of 80LBG-20GG/AG/*s*-HAp composite sheet, therefore, indicates that the crack deflection may occur during the fracturing process, showing the higher tensile stress. According to the FE-SEM micrographs shown in figure 8, the pores with sizes of approximately 1–2 μm are present in 80LBG-20GG/AG/*s*-HAp composite sheet, which may contribute to enhancing the ultimate tensile strength, regardless of the relative porosity as high as 85.4%.

## 3.3. Haemostasis properties of LBG–GG/AG/POs-Ca or *s*-HAp composite sheet

The tensile strength of the sheet is related to the staunching performance, and needed for the handling to cover the bleeding sites and haemostasis. Relating to the optimum composition for the balance between chemical composition and tensile strength, we have to further take into the consideration the performances of (i) tensile strength for the handling, (ii) large amount of blood uptakes, and (iii) excellent bioresorbability. In order to evaluate the blood staunching performance of the composite sheet for the application to the haemostasis materials, the uptake properties of the SBF were firstly examined with the results being shown in figure 10. The uptake amounts of SBF solution were: 1426 mass% (80LBG-20GG/AG/*s*-HAp composite sheet), greater than 750 mass% (80LBG-20GG/AG/POs-Ca composite sheet) and greater than 521 mass% (80LBG-20GG/AG composite sheet). Thus, the significant amount of SBF solution could be uptaken by 80LBG-20GG/AG/*s*-HAp composite sheet, rather than the case of 80LBG-20GG/AG/POs-Ca and 80LBG-20GG/AG composite sheets. The porosities of 80LBG-20GG/AG/*s*-HAp and 80LBG-20GG/AG/POs-Ca composite sheets were 81.0% and 85.4%, respectively. Regardless of the porosity of 80LBG-20GG/AG/*s*-HAp composite sheet being

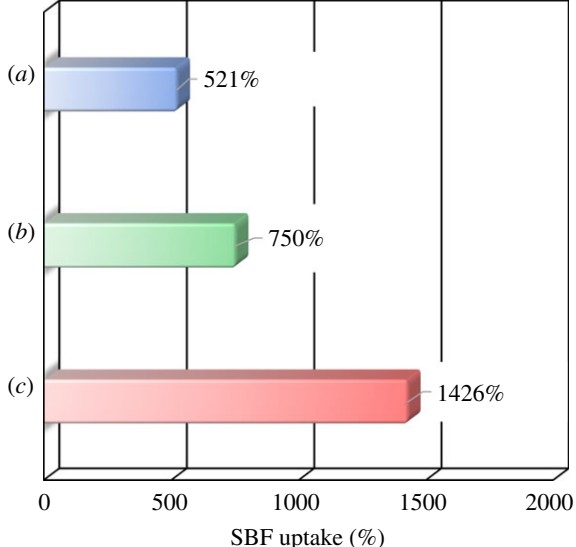

**Figure 10.** Amounts of SBF uptakes occurred with (*a*) 80LBG-20GG/AG composite sheet and (*b*) 80LBG-20GG/AG/POs-Ca composite sheet and (*c*) 80LBG-20GG/AG/*s*-HAp composite sheet fabricated by agitating sol at 20 000 r.p.m. (3 min), immersing in 3 mass% POs-Ca solution (R.T. for 24 h), hydrothermally treating at 100°C for 5 h (except for 80LBG-20GG/AG/POs-Ca composite sheet) and then freeze-drying at −50°C for 24 h. Immersion conditions: 37.0°C, 30 min.

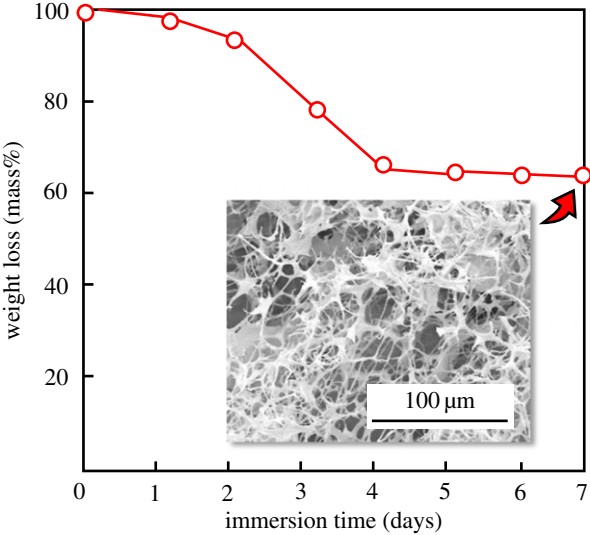

**Figure 11.** Mass change of 80LBG-20GG/AG/*s*-HAp composite sheet fabricated by agitating sol at 20 000 r.p.m. (3 min), immersing in 3 mass% POs-Ca solution (R.T. for 24 h), hydrothermally treating at 100°C for 5 h and then freeze-drying at −50°C for 24 h hydrothermally treated at 100°C for 5 h during the immersion in 0.1 mol dm$^{-3}$ acetic acid–sodium acetate at 37.0°C (pH = 5.5), together with FE-SEM micrographs.

lower than that of 80LBG-20GG/AG/POs-Ca composite sheet, the former uptake amount of SBF solution is rather larger than the latter case. As the microstructures of these composite sheets in figure 8 indicate, the pore sizes of 80LBG-20GG/AG/*s*-HAp composite sheet (1–2 µm) are rather uniform, compared with the case of 80LBG-20GG/AG/POs-Ca composite sheet (10–50 µm), and the former amount of closed pores that resists intruding the SBF solution into the composite sheet seems to be larger than the latter case.

The bioresorbability of 80LBG-20GG/AG/*s*-HAp composite sheet was examined by immersing it into 0.1 mol dm$^{-3}$ acetic acid–sodium acetate at 37.0°C (pH = 5.5). The mass loss during the immersion of 80LBG-20GG/AG/*s*-HAp composite in the 0.1 mol dm$^{-3}$ acetic acid–sodium acetate is shown in figure 11, together with typical FE-SEM micrographs. The mass loss occurred from 1.0 down to 0.65 with increasing time to 4 days, and gradual mass loss continued until 7 days. FE-SEM micrographs indicated that the

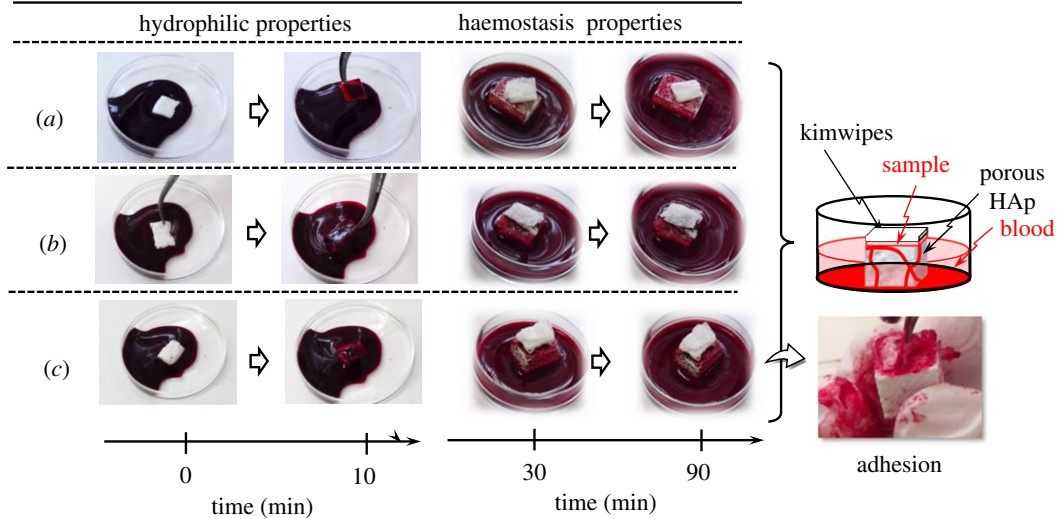

**Figure 12.** Changes in haemostasis time of (*a*) 80LBG-20GG/AG composite sheet, (*b*) 80LBG-20GG/AG/POs-Ca composite sheet and (*c*) 80LBG-20GG/AG/s-HAp composite sheet fabricated by agitating sol at 20 000 r.p.m. (3 min), immersing in 3 mass% POs-Ca solution (R.T. for 24 h), hydrothermally treating at 100°C for 5 h (except for 80LBG-20GG/AG/POs-Ca composite sheet), and then freeze-drying at −50°C for 24 h. Note that the blood-uptaken sheet was strongly adhered to the cubic HAp body.

noted absorption, as well as the pore size enlargement, was observed after the immersion for 7 days. As GG possesses extra galactose branch points, it seems to be soluble, rather than the case of LBG. Further, POs-Ca-derived components within s-HAp structure must be dissolved away during the immersion in the acetic acid–sodium acetate buffer solution. On the other hand, the structure of AG is reinforced by $Ca^{2+}$, which plays a role in the cross-linking of AG structure [22], indicating that the AG seems to resist the dissolution into the acid solution.

The haemostasis behaviours of 80LBG-20GG/AG composite sheet, 80LBG-20GG/AG/POs-Ca composite sheet and 80LBG-20GG/AG/s-HAp composite sheet examined using the human blood are shown in figure 12. When the composite sheets on porous HAp bodies were put in the human blood, they showed the excellent wettability with human blood. On the other hand, the haemostasis behaviour, which was evaluated by checking the time that wetted the composite sheet on the porous HAp body (see schematic illustration in the figure), indicated that the gradual blood uptake occurred within 30 min, i.e. the complete blood coagulation being caused by the platelet and coagulation factor. After the coagulation of the blood for 90 min, the composite sheets were strongly adhered to the porous HAp, showing the excellent blood staunching performance (see photograph). Regarding the 80LBG-20GG/AG/POs-Ca composite sheet, however, it was partially broken when pressed by fingers, in contrast to the case of 80LBG-20GG/AG/s-HAp composite sheet showing no changes in appearance even after the pressing by fingers and excellent adhesion to the porous HAp body (see photograph). Overall, 80LBG-20GG/AG/s-HAp composite sheet was found to be excellent haemostasis performance.

## 4. Conclusion

The fabrication conditions of bone-haemostasis materials were examined by using s-HAp derived from POs-Ca and natural plant-derived polymer (LBG, GG and AG). The results obtained were summarized as follows:

(i) The porous materials could be prepared by dissolving 2 mass% LBG/GG and 2 mass% sodium AG into the deionized water. The resulting sol was soaked into 3 mass% of POs-Ca solution for 24 h and was hydrothermally treated at 100°C for 5 h, followed by the freeze-drying at −50°C for 24 h.

(ii) The microscopic observation showed that the pore sizes were controlled in the range of 5–100 μm by changing the LBG/GG ratio and soaking time. The composite showed the noted uptake amount of SBF (1426%) at 37.0°C and also at the human blood.

(iii) The composites of HAp and natural plant-derived polymer are found to be promising materials for the bone haemostasis for its high solubility, haemostasis, uptake ability and adhesive properties.

**Table 1.** Reference values of X-ray diffraction for HAp: $2\theta$-values, $d$-values, relative intensities ($I_{rel}$) and corresponding indices (hkl) [13].

| $2\theta$ (°) | $d_{2\theta}$ (nm) | $I_{rel}$ | hkl |
|---|---|---|---|
| 10.84 | 0.816 | 8 | 100 |
| 16.82 | 0.527 | 3 | 101 |
| 18.80 | 0.472 | 1 | 110 |
| 21.76 | 0.408 | 6 | 200 |
| 22.84 | 0.389 | 6 | 111 |
| 25.35 | 0.351 | 3 | 201 |
| 25.85 | 0.344 | 44 | 002 |
| 28.14 | 0.317 | 9 | 102 |
| 28.93 | 0.308 | 15 | 210 |
| 31.76 | 0.282 | 100 | 211 |
| 32.16 | 0.278 | 59 | 112 |
| 32.89 | 0.272 | 59 | 300 |
| 34.03 | 0.263 | 24 | 202 |
| 35.45 | 0.253 | 5 | 301 |
| 39.17 | 0.2298 | 6 | 212 |
| 39.80 | 0.2263 | 21 | 310 |
| 40.40 | 0.2231 | 2 | 221 |
| 41.96 | 0.2151 | 6 | 311 |
| 42.30 | 0.2135 | 1 | 302 |
| 43.83 | 0.2063 | 5 | 113 |
| 44.38 | 0.2040 | 2 | 400 |
| 45.28 | 0.2000 | 4 | 203 |
| 46.69 | 0.1944 | 28 | 222 |
| 48.07 | 0.1891 | 12 | 312 |
| 48.59 | 0.1872 | 3 | 320 |
| 49.45 | 0.1842 | 30 | 213 |
| 50.49 | 0.1806 | 13 | 321 |
| 51.27 | 0.1781 | 9 | 410 |
| 52.07 | 0.1755 | 11 | 402 |
| 53.14 | 0.1722 | 15 | 004 |
| 54.41 | 0.1685 | 1 | 104 |
| 55.87 | 0.1644 | 5 | 322 |
| 57.10 | 0.1612 | 3 | 313 |
| 58.01 | 0.1588 | 2 | 501 |
| 58.27 | 0.1582 | 2 | 412 |
| 58.75 | 0.1570 | 1 | 330 |
| 59.91 | 0.1543 | 4 | 420 |

Ethics. The experiment regarding the human blood was conducted according to the ethics of the Department of Orthopedic Surgery at Toho University (Toho University Animal Care and User Committee).
Data accessibility. X-ray diffractometry data for the phase identification of the hydroxyapatite was accessed through the National Institute of Standards and Technology—Archived Certificates & Reports of Investigation (table 1 [13]). The standard SBF solution was prepared according to Kokubo's protocol.

Authors' contributions. Y.N. and K.I. made the present research plans. Y.N. prepared all samples for analysis, and collected/analysed the data. T.U. and Y.M. collected and analysed the data with human blood. Y.N. and K.I. interpreted the overall results and prepared the manuscript. All authors gave final approval for publication.

Competing interests. We declare we have no competing interests.

Funding. There are no funders to report for this submission.

Acknowledgements. No acknowledgements are needed in order to publish the research paper.

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
