## [Reviewer comments · Royal Society Open Science]

Review History

RSOS-181649.R0 (Original submission)

Review form: Reviewer 1 (Toshinori Okura)

Is the manuscript scientifically sound in its present form?

Yes

Are the interpretations and conclusions justified by the results?

Yes

Is the language acceptable?

Yes

Is it clear how to access all supporting data?

Not Applicable

Do you have any ethical concerns with this paper?

No

Have you any concerns about statistical analyses in this paper?

No

Recommendation?

Accept with minor revision (please list in comments)

Comments to the Author(s)

This paper describes new bone-hemostatic materials and it is very valuable.

Please correct the following careless mistakes before acceptance.

1. The number in the section "3.1 Fabrication of porous composite sheet" is 2.1.
2. The number in the section "3.2 Evaluation" is 2.2.
3. In section 3.3, it is better to unify unit notation of 1,426%, 750 mass% and 521%.

Also is there any correlation between the tensile strength of the composite and the stanching performance? What is the optimal composition when considering the balance between the two? Please comment if possible.

Review form: Reviewer 2

Is the manuscript scientifically sound in its present form?

Yes

Are the interpretations and conclusions justified by the results?

Yes

Is the language acceptable?

Yes

Is it clear how to access all supporting data?

Yes

Do you have any ethical concerns with this paper?

No

Have you any concerns about statistical analyses in this paper?

No

Recommendation?

Accept with minor revision (please list in comments)

Comments to the Author(s)

This paper describes the fabrication of novel bone hemostasis sheet by using sugar-containing hydroxyapatite and plant-derived polymer. As this paper contains the novel findings in the bioceramics field, it values to publish in the Royal Society Open Science. However, I think that the authors should revise the manuscript on the basis of the following comments.

- 1) What is "sugar-containing hydroxyapatite (s-HAp)"? Please explain it in detail. I think that there are many kinds of sugars in this study.
- 2) Please replace "bio-absorbability (bio-absorbable)" with "bioresorbability (bioresorbable)".
- 3) Page 2, line 16 from top:
The authors have used "human blood" in this work. The authors should make clear how did

obtain it.

4) Vertical axis in Figs. 7 and 11:

The authors should revise the “Mass lose / %” to “Weight loss / mass%”.

5) Page 5, line 7 from bottom:

The “SBF” is used in this work. The authors should explain the composition and preparation method in the section of 2. Materials and Methods.

Decision letter (RSOS-181649.R0)

09-Jan-2019

Dear Miss Noh

On behalf of the Editors, I am pleased to inform you that your Manuscript RSOS-181649 entitled "Fabrication of novel bone hemostasis sheet by using sugar-containing hydroxyapatite and plant-derived polymer" has been accepted for publication in Royal Society Open Science subject to minor revision in accordance with the referee suggestions. Please find the referees' comments at the end of this email.

The reviewers and handling editors have recommended publication, but also suggest some minor revisions to your manuscript. Therefore, I invite you to respond to the comments and revise your manuscript.

- Ethics statement

- Data accessibility

<http://datadryad.org/submit?journalID=RSOS&manu=RSOS-181649>

- Competing interests

- Authors' contributions

All submissions, other than those with a single author, must include an Authors' Contributions section which individually lists the specific contribution of each author. The list of Authors

should meet all of the following criteria; 1) substantial contributions to conception and design, or acquisition of data, or analysis and interpretation of data; 2) drafting the article or revising it critically for important intellectual content; and 3) final approval of the version to be published.

- Acknowledgements

- Funding statement

Because the schedule for publication is very tight, it is a condition of publication that you submit the revised version of your manuscript before 18-Jan-2019. Please note that the revision deadline will expire at 00.00am on this date. If you do not think you will be able to meet this date please let me know immediately.

- 1) A text file of the manuscript (tex, txt, rtf, docx or doc), references, tables (including captions) and figure captions. Do not upload a PDF as your "Main Document";

- 2) A separate electronic file of each figure (EPS or print-quality PDF preferred (either format should be produced directly from original creation package), or original software format);
- 3) Included a 100 word media summary of your paper when requested at submission. Please ensure you have entered correct contact details (email, institution and telephone) in your user account;
- 4) Included the raw data to support the claims made in your paper. You can either include your data as electronic supplementary material or upload to a repository and include the relevant doi within your manuscript. Make sure it is clear in your data accessibility statement how the data can be accessed;
- 5) All supplementary materials accompanying an accepted article will be treated as in their final form. Note that the Royal Society will neither edit nor typeset supplementary material and it will be hosted as provided. Please ensure that the supplementary material includes the paper details where possible (authors, article title, journal name).

on behalf of Dr Derek Abbott (Associate Editor) and Professor R. Kerry Rowe (Subject Editor)
openscience@royalsociety.org

Reviewer comments to Author:
Reviewer: 1

Comments to the Author(s)

This paper describes new bone-hemostatic materials and it is very valuable. Please correct the following careless mistakes before acceptance.

1. The number in the section "3.1 Fabrication of porous composite sheet" is 2.1.

2. The number in the section "3.2 Evaluation" is 2.2.
3. In section 3.3, it is better to unify unit notation of 1,426%, 750 mass% and 521%.
Also is there any correlation between the tensile strength of the composite and the stanching performance? What is the optimal composition when considering the balance between the two?
Please comment if possible.

Reviewer: 2

Comments to the Author(s)

This paper describes the fabrication of novel bone hemostasis sheet by using sugar-containing hydroxyapatite and plant-derived polymer. As this paper contains the novel findings in the bioceramics field, it values to publish in the Royal Society Open Science. However, I think that the authors should revise the manuscript on the basis of the following comments.

- 1) What is "sugar-containing hydroxyapatite (s-HAp)"? Please explain it in detail. I think that there are many kinds of sugars in this study.
- 2) Please replace "bio-absorbability (bio-absorbable)" with "bioresorbability (bioresorbable)".
- 3) Page 2, line 16 from top:
The authors have used "human blood" in this work. The authors should make clear how did obtain it.
- 4) Vertical axis in Figs. 7 and 11:
The authors should revise the "Mass lose / %" to "Weight loss / mass%".
- 5) Page 5, line 7 from bottom:
The "SBF" is used in this work. The authors should explain the composition and preparation method in the section of 2. Materials and Methods.

Author's Response to Decision Letter for (RSOS-181649.R0)

See Appendix A.

Decision letter (RSOS-181649.R1)

29-Mar-2019

Dear Miss Noh,

I am pleased to inform you that your manuscript entitled "Fabrication of novel bone hemostasis sheet by using sugar-containing hydroxyapatite and plant-derived polymer" is now accepted for publication in Royal Society Open Science.

Royal Society Open Science operates under a continuous publication model (<http://bit.ly/cpFAQ>). Your article will be published straight into the next open issue and this will be the final version of the paper. As such, it can be cited immediately by other researchers.

As the issue version of your paper will be the only version to be published I would advise you to check your proofs thoroughly as changes cannot be made once the paper is published.

on behalf of Dr Derek Abbott (Associate Editor) and Professor R. Kerry Rowe (Subject Editor)
openscience@royalsociety.org

Appendix A

March 25, 2019

Dear Sirs:

Thank you very much for the useful response to the following manuscript submitted to the Royal Society Open Science (Special issue for ISIEM2018).

Auhors: Y. NOH, T. UMEDA, Y. MUSHA and K. ITATANI

Title: Fabrication of novel bone hemostasis sheet by using sugar-containing hydroxyapatite and plant-derived polymer (Manuscript ID : RSOS-181649)

Please find our responses to the comments described below.

Sincerely,

Yeonjeong Noh
Department of Materials and Life Sciences
Sophia University

We have revised our manuscript on the basis of the reviewers' comments as follows:

Response to Reviewer: 1

1-1. The number in the section "3.1 Fabrication of porous composite sheet" is 2.1.

Response: Thank you very much for pointing it out. We have revised the section number.

1-2. The number in the section "3.2 Evaluation" is 2.2.

Response: Thank you very much for pointing it out. We have revised the section number.

1-3. In section 3.3, it is better to unify unit notation of 1,426%, 750 mass% and 521%.

Response: In section 3.3, we unify unit notation of the amounts of SBF uptakes as follows:
The uptake amounts of SBF solution were: 1,426 mass% (80LBG-20GG/AG/s-HAp composite sheet) > 750 mass% (80LBG-20GG/AG/POs-Ca composite sheet) > 521 mass% (80LBG-20GG/AG composite sheet).

1.4. Also is there any correlation between the tensile strength of the composite and the stanching performance? What is the optimal composition when considering the balance between the two? Please comment if possible.

Response: The tensile strength of the sheet is related to the stanching performance, and is needed for the handling to cover the bleeding sites and hemostasis. Relating to the optimum composition for the balance between chemical composition and tensile strength, we have to take into the considerations of the performances of (i) tensile strength for the

handling, (ii) large amount of blood uptakes and (iii) excellent bioresorbability. On the basis of such consideration, we added the explanation in the text as follows:

“The tensile strength of the sheet is related to the stanching performance, and needed for the handling to cover the bleeding sites and hemostasis. Relating to the optimum composition for the balance between chemical composition and tensile strength, we have to further take into the considerations of the performances of (i) tensile strength for the handling, (ii) large amount of blood uptakes and (iii) excellent bioresorbability.”

Response to Reviewer: 2

2-1. What is “sugar-containing hydroxyapatite (s-HAp)”? Please explain it in detail. I think that there are many kinds of sugars in this study.

Response: We added the details of sugar-containing hydroxyapatite (s-HAp) in Introduction. “Such POs-Ca may be partially hydrolysed to form a POs-Ca-derived organic compounds and hydroxyapatite ($\text{Ca}_{10}(\text{PO}_4)_6(\text{OH})_2$; HAp), *i.e.*, sugar-containing HAp (s-HAp) which helps assisting the bone regeneration [2,3].”

2. Please replace “bio-absorbability (bio-absorbable)” with “bioresorbability (bioresorbable)”.

Response: We replace the word “bio-absorbability (bio-absorbable)” with “bioresorbability (bioresorbable)”.

3. Page 2, line 16 from top: The authors have used “human blood” in this work. The authors should make clear how did obtain it.

Response: : We added the following sentence in the text as follows:

“The fresh human blood was collected from the adult male immediately before the test of hemostasis.”

4. Vertical axis in Figs. 7 and 11: The authors should revise the “Mass loss / %” to “Weight loss / mass%”.

Response: We revised the Figs. 7 and 11, on the basis of the reviewer’s suggestion.

5. Page 5, line 7 from bottom: The “SBF” is used in this work. The authors should explain the composition and preparation method in the section of 2. Materials and Methods.

Response: We added the composition and preparation method of SBF solution in the section 2.2 as follows:

“The specimen was immersed into the simulated body fluid (SBF) solution, and checked the uptake amount of SBF solution. The standard SBF solution was prepared by dissolving appropriate amounts of the chemicals in deionized water, *i.e.*, NaCl, NaHCO_3 , KCl, $\text{K}_2\text{HPO}_4 \cdot 3\text{H}_2\text{O}$, $\text{MgCl}_2 \cdot 6\text{H}_2\text{O}$, CaCl_2 , HCl, Na_2SO_4 and $(\text{CH}_2\text{OH})_3\text{CNH}_2$, using $1 \text{ mol} \cdot \text{dm}^{-3}$ of HCl. The concentrations of inorganic ions in the SBF solution were almost the same as the human blood plasma.”